# Smart Piezoelectric-Based Wearable System for Calorie Intake Estimation Using Machine Learning

**Ghulam Hussain** [1,*], **Bander Ali Saleh Al-rimy** [2,*], **Saddam Hussain** [3], **Abdullah M. Albarrak** [4], **Sultan Noman Qasem** [4] **and Zeeshan Ali** [5]

1 Electronic Engineering Department, Quaid-e-Awam University Larkana (Campus), Larkana 77150, Pakistan
2 School of Computing, Faculty of Engineering, Universiti Teknology Malaysia, Johor Bahru 81310, Malaysia
3 School of Electrical Engineering, UTM SKUDAI, Johor Bahru 81310, Malaysia; engineersaddamhussain@outlook.com
4 Computer Science Department, College of Computer and Information Sciences, Imam Mohammad Ibn Saud Islamic University (IMSIU), Riyadh 11432, Saudi Arabia; amsbarrak@imamu.edu.sa (A.M.A.); snmohammed@imamu.edu.sa (S.N.Q.)
5 Department of Telecommunication Engineering, Dawood University of Engineering & Technology, Karachi 74800, Pakistan; zeeshan.ali@duet.edu.pk
* Correspondence: engr.ghulam.hussain@quest.edu.pk (G.H.); bander@utm.my (B.A.S.A.-r.); Tel.: +92-312-087-4758 (G.H.)

**Abstract:** Eating an appropriate food volume, maintaining the required calorie count, and making good nutritional choices are key factors for reducing the risk of obesity, which has many consequences such as Osteoarthritis (OA) that affects the patient's knee. In this paper, we present a wearable sensor in the form of a necklace embedded with a piezoelectric sensor, that detects skin movement from the lower trachea while eating. In contrast to the previous state-of-the-art piezoelectric sensor-based system that used spectral features, our system fully exploits temporal amplitude-varying signals for optimal features, and thus classifies foods more accurately. Through evaluation of the frame length and the position of swallowing in the frame, we found the best performance was with a frame length of 30 samples (1.5 s), with swallowing located towards the end of the frame. This demonstrates that the chewing sequence carries important information for classification. Additionally, we present a new approach in which the weight of solid food can be estimated from the swallow count, and the calorie count of food can be calculated from their estimated weight. Our system based on a smartphone app helps users live healthily by providing them with real-time feedback about their ingested food types, volume, and calorie count.

**Keywords:** machine learning; calorie monitoring; food recognition; piezoelectric sensor; OA; wearable necklace

## 1. Introduction

Healthy eating and calorie balancing are key factors for living a healthy life with a reduced risk of chronic diseases such as Knee Osteoarthritis (OA), cancers, heart disease, cirrhosis, and diabetes [1]. In OA, the pressure coming from the increased weight causes degradation of joint cartilage and the underlying bone. This, OA consequently causes stiffness and severe pain to the patient.

There have been attempts to design an automated food monitoring system that measures caloric intake by classifying types of food and estimating ingested amounts. Different sensors such as piezoelectric sensors, microphone, accelerometer, gyroscope, camera, weight scale, electromyograph, and textile pressure sensor have been employed to automatically monitor food intake.

Among these sensors, microphones have been widely used because they are more accurate than other sensors [2–9]. Microphone-based wearables collect acoustic signals non-invasively while the wearer is eating; however, surrounding noise affects the performance

and decreases the accuracy of food classification in real environments. Furthermore, these wearables have the disadvantages of low user acceptance for long-term usage [4] and low comfortability because they exert a large force against the neck [2,3]. On the other hand, the necklace-embedded piezoelectric sensor, also known as vibration sensor, generates distinct patterns for the chewing sequence and swallowing activity by detecting skin motion of the neck and jaw. The employed sensors are not only less sensitive to surrounding noise but also provide better user comfortability than microphone-based wearables.

Nowadays, the severity of chronic diseases has increased such that they become difficult to cure with traditional medicine. Medicine may cause several severe side effects while curing a disease. Thus, several other diseases attack the human body when using medicine for cure. Chronic diseases can be prevented if timely preventive steps are taken. Automated wearable monitoring systems are essential for assisting users in following preventive measures. Thus, it motivated us to design a novel food-intake monitoring system that can enable users to control their food intake and avoid deadly chronic diseases.

In this work, we present a smartphone application-based nutrition monitoring wearable system that classifies food categories and estimates caloric intake by taking advantage of the robust features of a piezoelectric embedded necklace. Our system classifies foods into different categories based on chewing and swallowing patterns and estimates the volume and calories of food ingested based on the number of swallows. As shown in Figure 1, our smartphone application sends statistics about caloric intake, suggests the next mealtime, and recommends physical exercise. The users can develop good healthy dietary behavior by following the recommended suggestions.

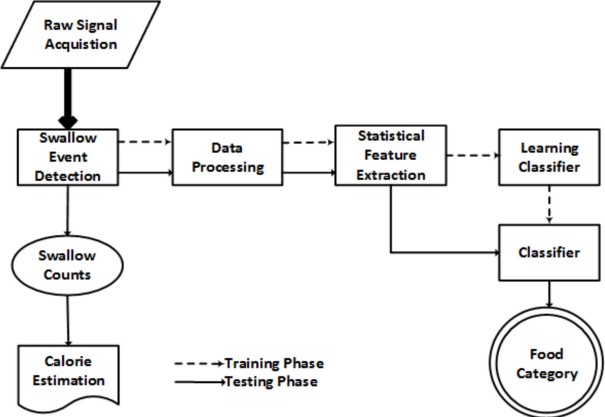

**Figure 1.** An overview of food intake monitoring system.

There are three components of our proposed food intake system. parts. First, we have achieved a higher classification accuracy than previous nutrition monitoring systems [2–7,9–28]. We have demonstrated that accurately processed raw temporal data along with an important set of optimal feature selection, enables the proposed system to outperform other state-of-the-art necklace systems [10,11] that use spectral features to recognize foods. We have observed that a temporal frame of 30 samples (1.5 s) containing a sequence of chewing events combined with a successive swallowing event forms a distinct pattern. The distinct temporal patterns in the frames generate essential effective features to associate the pattern with a relevant food category. Our proposed system has attained f-measures of 94.2%, 93.7%, and 95.1% using the k-nearest neighbor, support vector machine, and random forest, respectively. The second important aspect of this algorithm is its high estimation rate for the weight of solid foods, the main contributors to caloric intake. To our knowledge, this is first time that the weight of consumed food has been estimated from the number of swallows. This method has achieved comparatively better recognition accuracy than existing systems, with the additional advantage of avoiding restrictive or immovable models such as tables [26,27]. The estimated weight of food is converted into caloric intake with sufficient accuracy to make users aware of their

excessive caloric intake. Third, we built a smartphone application that provides a real-time notification about the number of swallows, estimated food volume, and number of calories during each meal.

This paper is organized as follows. Section 2 describes related work, focusing on food intake monitoring with piezoelectric sensors. Section 3 presents the raw data acquisition using our necklace embedded with the piezoelectric sensor and the data processing required to extract the discriminating features. Statistical feature extraction, selection, and food intake classification are described in Section 4. In Section 5, the experimental setup is discussed. We evaluate the system for its food recognition, weight approximation, and calorie estimation in Section 6, and provide a conclusion in the last section.

## 2. Literature Review

There has been significant effort by various researchers in the wearable healthcare field to design a wearable sensor to monitor and measure dietary intake. Researchers have designed various automated non-invasive food intake recognition systems using different sensors such as microphones [2,4–9,17], accelerometers [14], cameras [18], gyroscopes [21–24], textile pressure sensors [27], strain gauges [28], piezoelectric sensors [3,10,11,13,19], and orientation sensors [15,29].

Recently, to estimate food quantity for the purpose of health monitoring, smart-watches [4], bite counters [21–24], video recordings [18], and smart dining tables [26,27] have been used. Smartwatch-based eating-behavior evaluation [4] has been proposed as a new modality for food intake monitoring. This system uses the microphone of the smart-watch, and has been used by many researchers in different configurations [2,4–9,17]. The smartwatch-based system has limitations such as the effects of surrounding environmental noise and the inability to estimate the quantity of food. In addition to microphone-based systems, other sensor-based systems have many limitations compared to piezoelectric sensor-based systems, such as their fixed environmental settings [26,27] or the limited scope of bite counters [21–24].

We have used a necklace embedded with a piezoelectric sensor to track or transform the movement of neck skin into a suitable voltage. As each class of food is different to another, so bites of different foods cause different amounts of movement in the neck skin. Hence, the piezoelectric-sensor embedded necklace was chosen to accurately represent different skin movements with distinct patterns of voltages. The dietary intake recognition systems based on piezoelectric sensors are summarized in Table 1. N. Alshurafa et al. developed a wearable sensor system that consisted of a necklace to monitor nutrition intake [10,11]. The necklace was embedded with a piezoelectric sensor to detect skin motion in the lower trachea during ingestion. The authors' method of food classification was applied on a limited number of binary categories such as liquid and solid, cold and hot, and soft and hard. They developed a classification model using statistical features extracted from a spectrogram. The authors used a long sample window and an improper swallowing position that obscured the swallowing characteristics. Accordingly, their spectrogram did not exhibit salient statistical features over time. Thus, this model performed only moderately at the cost of high complexity and extra computation of the spectrogram.

H. Kalantarian et al. investigated the concept of food intake monitoring using a low-cost piezoelectric-sensor embedded in a necklace [13,19]. The sensor generated a signal with a distinct pattern that enabled the system to recognize water, sandwiches, and potato chips. They reported food classification accuracies of 81.4%, 84.5%, and 85.3% for water, sandwiches, and chips, respectively. The authors' methodology obtained a better performance than [10,11] by processing the temporal data and avoiding the computationally complex short-time Fourier transform (STFT).

**Table 1.** A summary of necklace-based intake monitoring systems.

| Sensor Type | Description | Sensor Form | Food Classes (Number of Subjects) | Accuracy (%) |
|---|---|---|---|---|
| Piezoelectric | Refs. [10,11] presented a unique wearable system to detect neck skin movement caused by ingestion. | Necklace. | Water, hot tea, patty, chocolate, and nuts (20). | 87% [10] and 90% [11] for solid and liquid. 90% for hot and cold. 80% for solids. |
| | A smartphone application-based nutrition-intake monitoring system [19] consisting of a necklace similar to [10,11] estimated meal volumes. | Necklace. | Water, sandwich, and chips (10). | 85.3% for chips, 84.5% for sandwich and 81.4% for water. |
| Piezoelectric and Accelerometers | The wearable system of [10,11,19] was improved by the addition of an accelerometer in [13] to decrease detection of false positive swallows. | Necklace. | Sandwich, chips, and water (30). | 85.3% for chips, 84.5% for sandwich and 81.4% for water. |
| Piezoelectric or Microphones | The performance of microphone and piezoelectric for swallow detection were compared in [3], when used separately. | Necklace and throat microphone. | Chips, sandwich, and water (10). Mixed nuts, patty, and two small chocolates (20). | 91.3% and 88.5% (microphone). 75.3% and 79.4% (necklace). |

H. Kalantarian et al. compared a piezoelectric sensor and a microphone, the two most promising intake measuring sensors for automated dietary intake monitoring systems [3]. They obtained better food classification with the microphone than with the piezoelectric sensor. They applied a long sample window that failed to accurately process the data and chose an unsuitable position for the swallowing event. The authors obtained features from a spectrogram as they were inspired by [10,11]. The spectrogram-based signal-processing technique was better than matching pursuit and scalogram-based Gabor wavelets.

However, as mentioned earlier, efficient features can be obtained from the temporal signal without the use of a spectrogram if a proper frame length and a suitable swallowing position are selected. In our work, we processed data and extracted features from the time–domain signal as reported in [13,19].

## 3. Acquisition of Raw Data

A necklace embedded with a piezoelectric sensor is deployed to sense and record ingestion patterns comprised of chewing sequences and swallow events. The smartphone application (App) establishes coordination between the Simblee microcontroller and the necklace. The sampling frequency of the App is 20 samples/second. The users manually insert the per-gram caloric value for each specific food category, as displayed in Figure 2.

### 3.1. A Piezoelectric-Sensor Embedded Necklace

Piezoelectric-sensor embedded necklaces are designed in two different configurations: pendants and sports bands [13]. Unlike [4,10,11,19], we selected the stretchable sports-band design. The sports-band design can easily be stretched to the users' neck, so people of different body structures can wear them without any extra effort, and hence it has better usability. As stated in [13], the stretchable necklace has the enormous advantages of comfortability and stability. The sports band is highly preferred in data collection, clinical environments, and algorithm evolution. The pendant [10,11,19], the second necklace design, has the problem of leaving the neck-skin position during motions such as vigorous head-turning, walking, or running. Therefore, a piezoelectric sensor embedded in a stretchable sports-band necklace is selected as shown in Figure 3. The necklace collects accurate experimental data by translating throat skin motion into unique voltage pattens using a single vibration sensor.

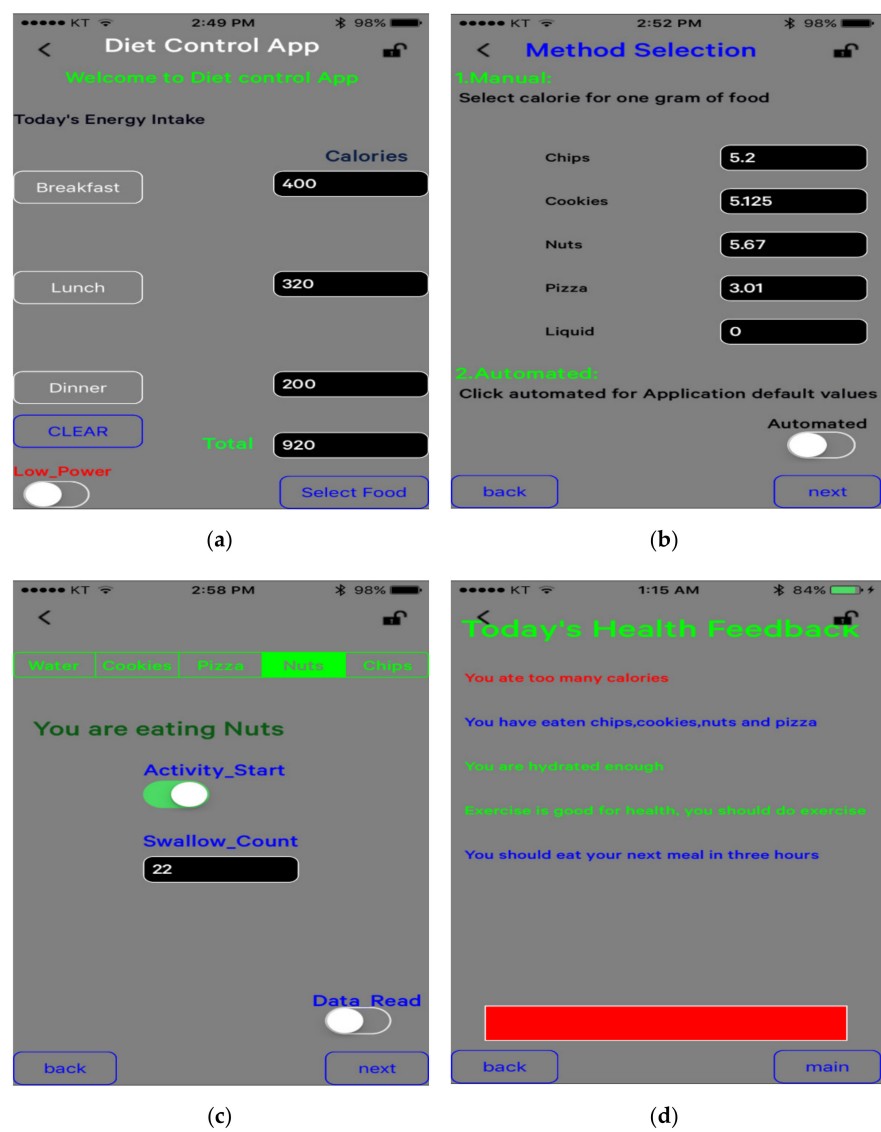

**Figure 2.** Smartphone application for food intake monitoring: (**a**) Caloric intake statistics; (**b**) Calorie method selection; (**c**) Swallow detection for each food category; (**d**) Health feedback for friendly suggestions.

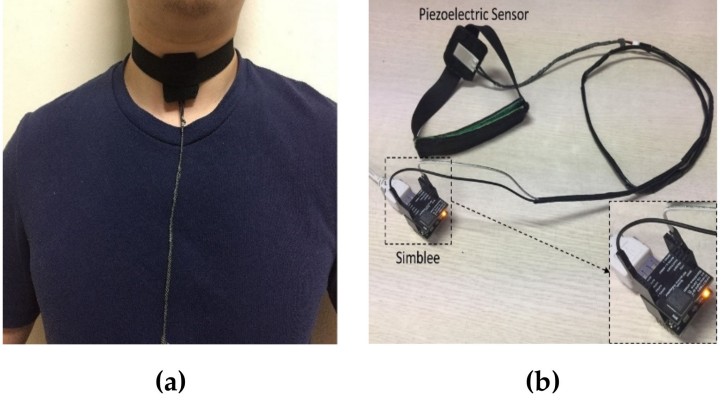

**Figure 3.** Necklace-type wearable system: (**a**) Subject wearing a necklace; (**b**) Necklace connected to Simblee.

The piezoelectric sensor generates a distinct pattern of output voltages according to the amount of pressure the neck skin applies against the sensor. During eating, neck muscles contract and relax in a way that they apply force on the vibration sensor. Hence, the sensor generates a distinct temporal pattern of voltages that represents chewing and swallowing events, as depicted in Figure 4. The piezoelectric-sensor embedded necklace is attached with the general-purpose input/output (GPIO) pin of the Simblee microcontroller, which has built-in analogue to digital converters (ADCs) convert the analogue signal into the digital signal. The sensor without an additional mass carries a sensitivity of 50 mV/g and resonates with magnitude of 1.4 V/g [29]. The smartphone application is the center of coordination between the sensor and the Simblee [30]. The smartphone application enables the sensor to monitor the user's skin movement with the coordination of the Simblee. Thereafter, the sensor continuously measures skin motion until the intake activity is ended.

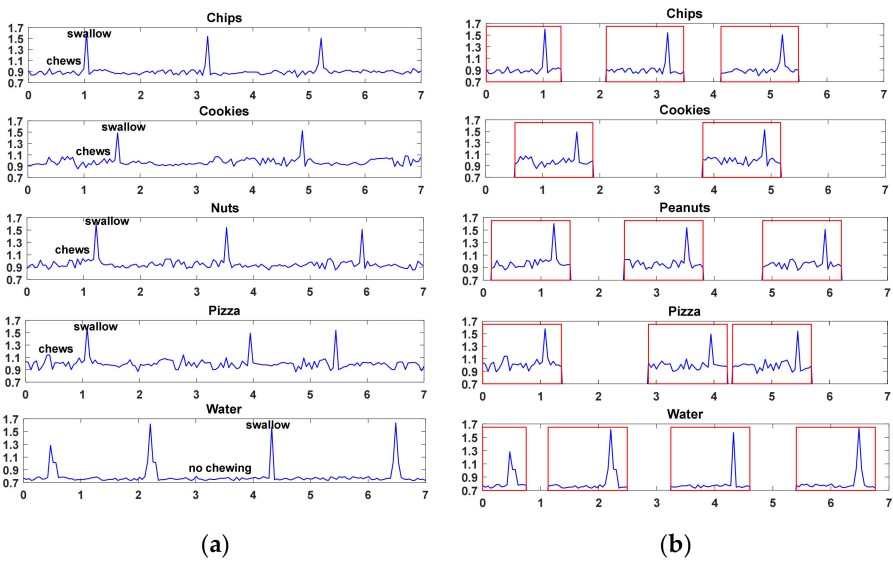

**Figure 4.** Food ingestion pattern waveform: (**a**) The eating pattern of all the ingested food categories; (**b**) Extracted frames from the ingestion pattern of foods are enclosed by rectangles.

Thus, the hardware system receives a temporal amplitude-varying voltage-signal, representing the ingestion pattern collected at 20 samples per second. The Simblee, a microcontroller, is more compact and smaller in size than the food monitoring system in [2]; thus, it can be carried easily in real-life settings, as depicted in Figure 3.

The smartphone application played an essential role in the implementation of the food intake monitoring system. The application acts as an interface between the intake monitoring system and the user. The application facilitates the experiment by providing different daily meal options, calorie calculation methods, food types, and caloric intake for each meal, and makes suggestions to the user based on their hydration level and caloric intake (see screenshots of the application in Figure 2). In contrast to [11,13], the user can extend the battery power of the intake monitoring system by switching the active mode of the Simblee microcontroller to ultra-low power (sleep) [31].

The weight of each food is estimated separately, based on the swallow count. The calories for each meal of the day are calculated by Equation (1), which aggregates the calories of all the ingested food types. The calories for each food type are calculated as the 1-g caloric weight, $c_i$ (coefficient), multiplied by the estimated food weight, $food_i$ (swallow counts). Thereafter, the calculated calories for all ingested foods are aggregated to yield the calories for a particular meal of the day. There are two methods of caloric coefficient insertion: user-defined and application-default. User-defined caloric coefficients are entered by the user, while application-default caloric coefficients are pre-defined or pre-programmed based on average values in order to generalize varieties of the same food.

The calorie estimation performance of these two methods are compared in Section 6.2. The caloric intake is simply estimated with Equation (1).

$$\text{caloric intake} = \sum_i c_i food_i \tag{1}$$

In this work, we considered five food types: chips, cookies, nuts, pizza, and water. As water was consumed in our experiment and does not contain any calories, the caloric coefficient for water is 0. For other foods, coefficients are entered by the user or left as the application default. Default values are calculated as the average number of calories from different varieties of the same food. Though the default values estimate calories less accurately for some foods, as discussed in Section 6.2, they also generalize the calorie equation well over different varieties of foods. In our experiment, foods were consumed in sequences of bites weighing about 1 g, corresponding to 3.01 calories for pizza, 5.67 for nuts, 5.2 for chips, and 5.125 for cookies for the estimated 1-g bolus ingested during each swallow.

### 3.2. Data Processing

The signal generated by the necklace is denoised to remove noisy data and to retain important ingestion events. It is important to clean the sensor signal prior to feature extraction and the development of the statistical model. Prior researchers [9,14,17] segmented the ingestion activity into three sequential phases: oral food preparation phase (food is chewed and turned into a viscous bolus), pharyngeal phase (bolus traveling through the pharynx and upper esophageal sphincter), and the esophageal phase (bolus enters the stomach via the esophagus). We combined the esophageal and pharyngeal phases to denote swallowing, and the oral preparation phase to represent chewing. The main objective is to obtain the signals representing the users' eating patterns for different foods. When food enters the mouth, different oral receptors are activated and carry sensory information about material properties to the brain. The most important stimuli are related to food texture (crispness, hardness, dryness, size, and shape) and flavor. The users continuously adjust these stimuli to efficiently break food into a bolus that can easily be swallowed. We have applied an equal weight to both actions: chewing and swallowing. As shown in Figure 4, the series of minor peaks represent chewing patterns and the major peaks with large amplitude represent swallowing patterns.

It is observed while ingesting different foods that users exert different amounts of force during each attempt at food breakdown because foods vary in hardness, crispiness, and texture. Thus, the forces exerted to breakdown foods with different levels of hardness generate distinct chewing patterns prior to each swallow. By exploiting the distinct ingestion patterns for foods with different textures, a series of chewing events are combined with a swallowing event, which forms data of ingestion patterns. In contrast to [10,11], we employ frames to borderline the desired ingestion pattern, ultimately enabling the efficient extraction of the efficient discriminating features. The frames enclosed in rectangular boxes are shown in Figure 4b. The proper window length for the samples and a suitable swallowing position in the frame should be chosen for each frame representation of the raw data, so that the features can be extracted successfully, and an efficient and accurate model can be built.

The raw signals represent the chewing and swallowing patterns for different foods. The acquired sensor signal is preprocessed to capture useful information from an event and thus to classify it correctly. Similar to the naive window selection method of [10], our method employs different non-overlapping window lengths to process the raw sensor signals and divide the signals into frames. An optimal window length was selected based on the performance of the classifier. As shown in Figure 5, window lengths of 20 samples (1 s) and 30 samples (1.5 s) are most suitable, based on an accuracy comparison of the models over different windows of samples. Previous research [13] has also indicated that a window length of 1 s (corresponding to 20 samples) is suitable for the correct classification of eating patterns from a piezoelectric sensor. However, we have selected a window length

of 1.5 s (30 samples) because of its slightly higher classifier performance. Four to five samples are used to represent a swallowing event, but a suitable count of samples are required to denote the chewing sequence; as the number of samples is important for the representation of the ingestion events and can affect the performance of the model if the number of samples in a window is too large or too small.

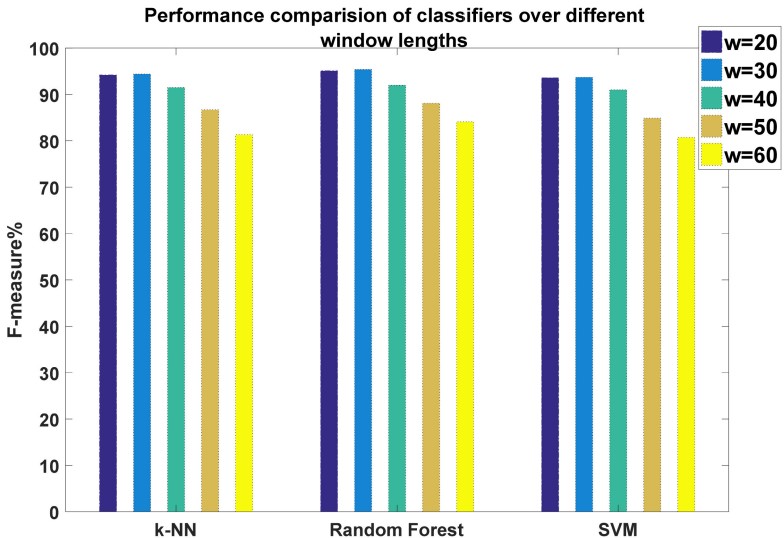

**Figure 5.** This figure demonstrates the impact of the window length over the classifiers' performance. The 30-sample (1.5-s) window is most suitable as evaluated on the performance of the classifiers.

The window of 30 samples (1.5 s) almost agrees with published data indicating that 500 ms is a sufficient window length to represent chewing strokes [32], and that 20 samples are adequate to represent chewing and swallowing [3,13] for food classification. The resulting frame for the signal after multiplication is shown in Figure 6a. The frame boundaries are indicated by two adjacent dashed red vertical lines. We then apply drop-out technique over all the frames or segments of the signal to search for informative frames containing the swallowing peak and to remove the noisy frames. Frames are selected or dropped out based on the swallowing peak. Frames containing the swallowing event peak are moved backward or forward to capture the preceding sequence of chewing events through which the food was transformed into a bolus. Food in the form of a viscous bolus is the final state prior to swallow stage. It has been observed that users do not enter extra food into mouth until swallowing the bolus; thus, chewing and swallowing events of the same intake cycle are associated with same food category. Thus, the frames contain 23 chewing samples (pre-swallow samples) and six post-swallow samples in order to cover the complete intake cycle. The noisy and uninformative frames are discarded, as depicted by the dropped-out section in Figure 6a.

N. Alshurafa et al. used a long sample window for data representation and spectrogram generation [11], but failed to extract distinct statistical features for food classification. The data processing technique of our algorithm is compared with theirs in Figure 6. As discussed in Section 6.1, their algorithm is much less accurate than ours for food classification because the authors selected a long pattern of chewing, which obscured solid swallows, especially. Moreover, they applied STFT to the collected data, which further complicated the data and led to the extraction of indistinct statistical features. The authors built a predictive model based on those inefficient features. However, they did capture swallowing peaks well, which may be advantageous to people with eating disorders.

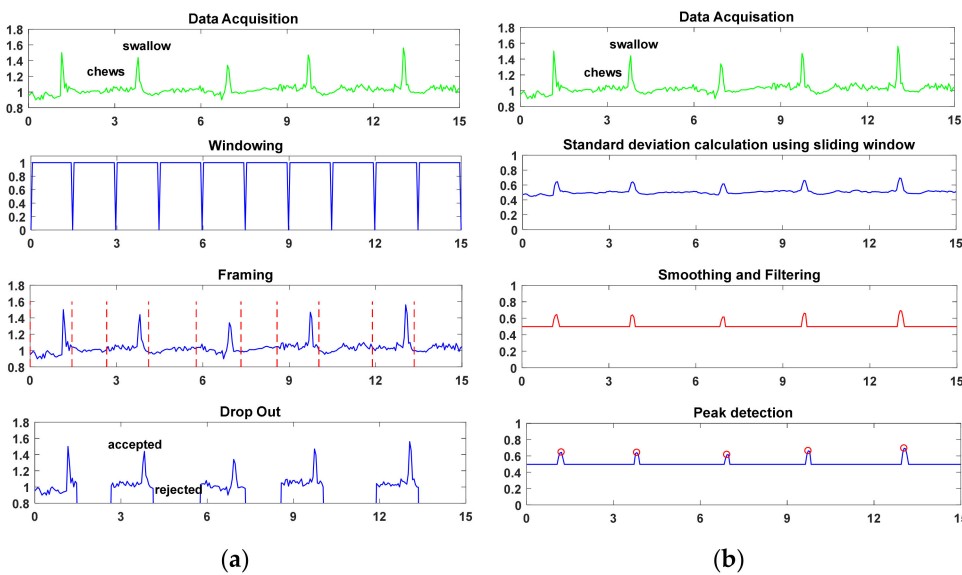

**Figure 6.** Comparison of data processing techniques: (**a**) Proposed data processing technique; (**b**) Previous work data processing technique [4,7].

We have positioned the swallow at the 4/5 of the frame length because this allows the window to cover the prior chewing pattern along with each succeeding swallow, in contrast to other positions of swallowing in the window. When swallowing is positioned elsewhere in the window, the waveforms of other processes are included, such as breathing, which always follows swallowing [33].

For both 20- and 30-sample frames, the best classification performance is achieved when swallowing is positioned towards the end of the frame. The general trend in Table 2 illustrates that the recognition performances of k-NN, random forest, and SVM increase when more chewing samples are included in the frame. This indicates that the chewing portion includes important information for classification. Therefore, in this assessment, we have obtained the best performance by using the random forest for a 30-sample frame with swallowing positioned at the 4/5 of the frame length. Frames of 30 samples perform slightly better than frames of 20 samples. Thus, we have chosen 30-sample frames after exploring both 20- and 30-sample frames with different swallowing positions in order to find the optimal raw signal for effective feature extraction. The results are given in Table 2.

**Table 2.** Classifier performance with 20- and 30-sample windows.

| Method Data Collection | Classification | k-NN (k = 3) | Random Forest Classifier | SVM |
|---|---|---|---|---|
| 20 samples (1 s) with different swallow positions | 1/4 | 92.5 | 93.2 | 92.3 |
|  | 2/4 | 93.7 | 93.1 | 92.0 |
|  | 3/4 | 94.2 | 95.1 | 93.7 |
| 30 samples (1.5 s) with different swallow positions | 1/5 | 93.7 | 93.8 | 92.9 |
|  | 2/5 | 94.1 | 94.3 | 93.0 |
|  | 3/5 | 94.2 | 94.6 | 93.3 |
|  | 4/5 | 94.4 | 95.4 | 93.7 |

As shown in Figure 4, the waveform after swallowing becomes flat during the breathing or silent phase, which is undetectable by the employed piezoelectric sensor, and lowers the accuracy of the system if it is included in the sample window (i.e., when the position of swallowing in the window is changed). Swallow positioning at the end of the window, in addition to performing well also agrees with the generalization assumption of [7], in which each swallow was combined with the preceding chewing pattern to form one intake

cycle before a new piece of food was ingested. In contrast, in the most similar work given by [10,11], swallowing was set in the middle of the window, so the window included silent phases along with related eating patterns; thus, the algorithm could not attain a high accuracy. Discriminating features are obtained from informative frames carrying information about the swallow event and the prior chewing sequence.

## 4. Feature Selection and Food Classification

A set of optimal features are chosen based on their predictive capability to associate ingestion patterns to a food category. Twelve statistical features (the arithmetic mean, standard deviation, first eigenvalue, harmonic mean, interquartile range, kurtosis, median, maximum, range, skewness, geometric mean, and z-score mean) are computed from processed or informative frames in order to classify different food classes by exploiting variations in the pattern of chewing sequence and swallowing event for each food. We have applied a heuristic algorithm to select efficient and non-redundant features to enable the classifier to distinguish various food categories.

There are various filtering algorithms that can be employed to select important features and organize them according to rank. Here, RELIEFF [34], an instance-based method, has assigned the relevance score to all extracted features. The score or weight of the features denotes their ability to distinguish the food categories. The features are organized in order of descending score, and low ranked features are removed. Accordingly, ten high-scoring features form the finalized set of optimal features to build the food classifiers.

*RELIEFF*

RELIEFF, a heuristic-based feature-filtering algorithm, selects a set of optimal features by assigning weight to the features. The algorithm randomly chooses near instances and applies the features over the chosen instances. The filtering method assigns a high weight or score to features that exhibit the ability to distinguish among near instances. The algorithm computes the difference between two instances based on the attributes or features using **function diff** (Attribute, instance1, instance2)/n. The features differ by discrete value, either 0 (equal values) or 1 (different values), whereas features in a continuous domain differ by the actual difference normalized to the interval [0, 1]. The weight-scores denote the strength of the feature. The attribute score is updated based on the notion that informative features should have the same score for the instances belonging to same class and should have different values for instances belonging to different categories. The RELIEFF method computes the feature weight $W[A]$ by Equation (2).

$$W[A] = P(\text{value of A}|\text{different class nearest instance}) - \\ P(\text{value of A}|\text{same class nearest instance}) \tag{2}$$

---
**Algorithm 1** The feature selecting RELIEF algorithm

---
*1: Set all weights W [A] =0.0*
*2: for i: =1: n (number of random instances) do*
*3:     begin*
*4:     Randomly select an instance R*
*5:     Find nearest hit H (same class) & nearest miss M (different class).*
*6:     for A: =1: all attributes do*
*7:   $W[A] := W[A] - \frac{diff(A,R,H)}{n} + \frac{diff(A,R,M)}{n}$*
*8:   end for*
*9: end for*

---

The algorithm is applied to the twelve features to compute their discriminating powers, which are shown in Figure 7. The rank of mean of standardized z scores and kurtosis is at the bottom of the important features list as the feature selecting algorithm is applied. It shows that both features at the bottom carry redundant and irrelevant information about ingestion patterns. Therefore, both features are eliminated to reduce the the model

complexity and improve its accuracy. The classifiers are built on the remaining optimal set of features computed from the ingestion patterns of different foods.

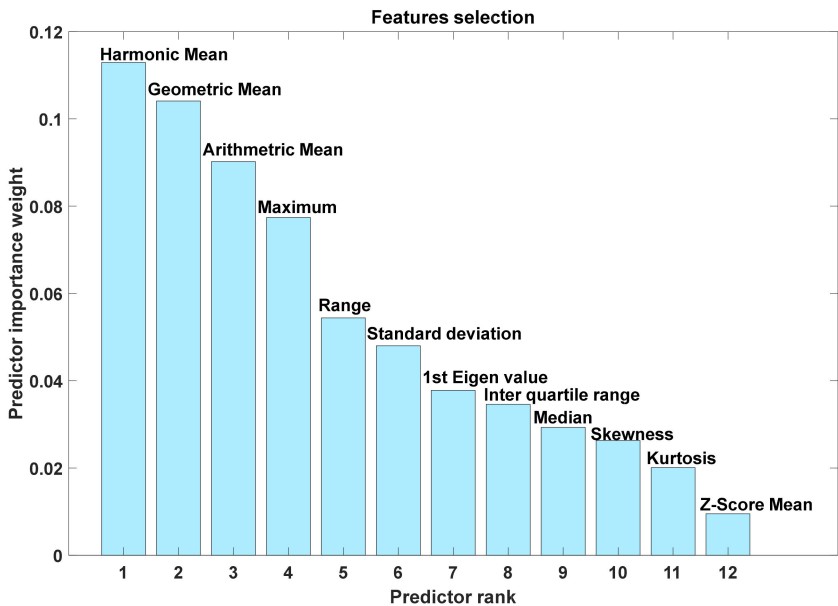

**Figure 7.** Important features in descending rank.

## 5. Experiment

We recruited eighteen graduate students at Sungkyunkwan University, eleven males and nine females, who had no disorder related to eating. The users agreed by signing consent forms prior to the experiment, and their personal data was kept private in accordance with the declaration of Helsinki. The subjects were subsequently invited for an individual recording in three sessions. Cheese pizza, chocolate chip cookies, peanuts, Pringles chips, and water were used for food intake testing of the proposed wearable nutrition-monitoring system. Participants were directed to eat only one food at a time and were prohibited from eating other foods in between these foods, so that the experimental data for each food category could be saved with the correct label by memory.

The users were instructed to eat a bite of food weighing about 1 g so that each bolus would weigh 1 g before it was swallowed. Swallow counts were used to estimate the weights of the ingested foods and the calories consumed, and to facilitate food recognition. Each chip and peanut weighed about 1 g. One cookie weighed about 6 g. The pizza was cut into 10-g slices, and a plastic cup containing 10 mL of water was used for each serving. A 1-g estimated bite was demonstrated in front of each participant 30 min before the start of the experiment. Participants were directed to consume a whole chip or one peanut, and to perform six bites for each cookie, 10 bites for one slice of pizza, and 10 sips for each cup of water.

At the beginning of the experiment, the necklace was calibrated around the user's neck and the user was handed a mobile set displaying the diet and caloric intake monitoring application. When the user turned on the switch displayed on the screen, a wake-up signal was given to the Simblee microcontroller board to be ready for intake-activity monitoring, and the system started acquiring data from the piezoelectric sensor after 2 s. The two-second delay in acquiring the sensor signal was selected to allow the user time to pick up the glass of water or take the first bite of food after switching on the food-intake activity.

Data collection for the ingestion activity was undertaken in a real environment; for example, other researchers sitting in the laboratory were allowed to talk, and the sound of the door opening and closing was permitted. The necklace-embedded piezoelectric sensor (the data-collecting source) is not influenced by surrounding noise; its waveform changes

when the neck skin exerts force. In contrast, microphone-based food monitoring systems exhibit lower recognition performance in real environments due to surrounding noise.

## 6. Result and Discussion

The performance of our proposed method is compared to most relevant previous studies as shown in Table 3. The performance and limitation of relevant studies and our method are described. It is evident that our method has outperformed several relevant studies with higher accuracy, better usability, and low limitations. However, there are a few studies based on advanced deep-learning models that extract features and perform recognition together [35,36]. We have not used computationally complex deep learning models as we did not want to improve classification accuracy with the cost of high computation but preferred simple machine learning models. The simple model used attained better classification accuracy and can be integrated in wearable devices to perform its operation in real-time.

**Table 3.** The performance comparison between the proposed study and previous related studies.

| Description | Performance | Limitation |
|---|---|---|
| A wearable system presented in [10,11] to detect neck skin movement caused by ingestion. | Their method achieved maximum accuracy of 90% for a small number of categories. | Binary classification |
| Nutrition intake monitoring system [19] consisting of a necklace similar to [10,11] estimated meal volumes. | The system attained average accuracy of 83%. | Low food categories |
| The authors improved wearable system of [10,11], and [19] by the addition of an accelerometer to decrease detection of false positive swallows [13]. | The system attained average accuracy of 83%. | Low food categories |
| The authors compared the performance of a microphone and piezoelectric sensor for swallow detection [3], when used separately. | The microphone-based system exhibited about 10% higher performance than necklace-based system. | Low recognition accuracy for fair count of food classes. |
| Our proposed method based on piezoelectric sensor recognized suitable count of food classes by exploiting an accurate flexible sensor with better data processing technique | Our method achieved average recognition performance of 94% for five food classes | Low food classes |

A detailed performance comparison of the current study and previous studies is discussed in following sections.

### 6.1. Food Classification

Our method attained a high recognition rate for five food types (chips, cookies, nuts, pizza, and water) when 30 samples (1.5 s) were used. Our method is compared to the most similar published method [10,11,13,19] in Figure 8, which clearly demonstrates that our method performs better than the previous method in recognizing all food types. Random forest, k-NN, and SVM are used to assess the effectiveness and efficiency of the two algorithms. The performance of the classifiers is evaluated based on precision, recall and the f-measure. The previous methods [10,11] exhibited a moderate recognition rate because of their large sample window. We think that large windows average out short-duration swallowing events, and STFT cannot produce distinct statistical features.

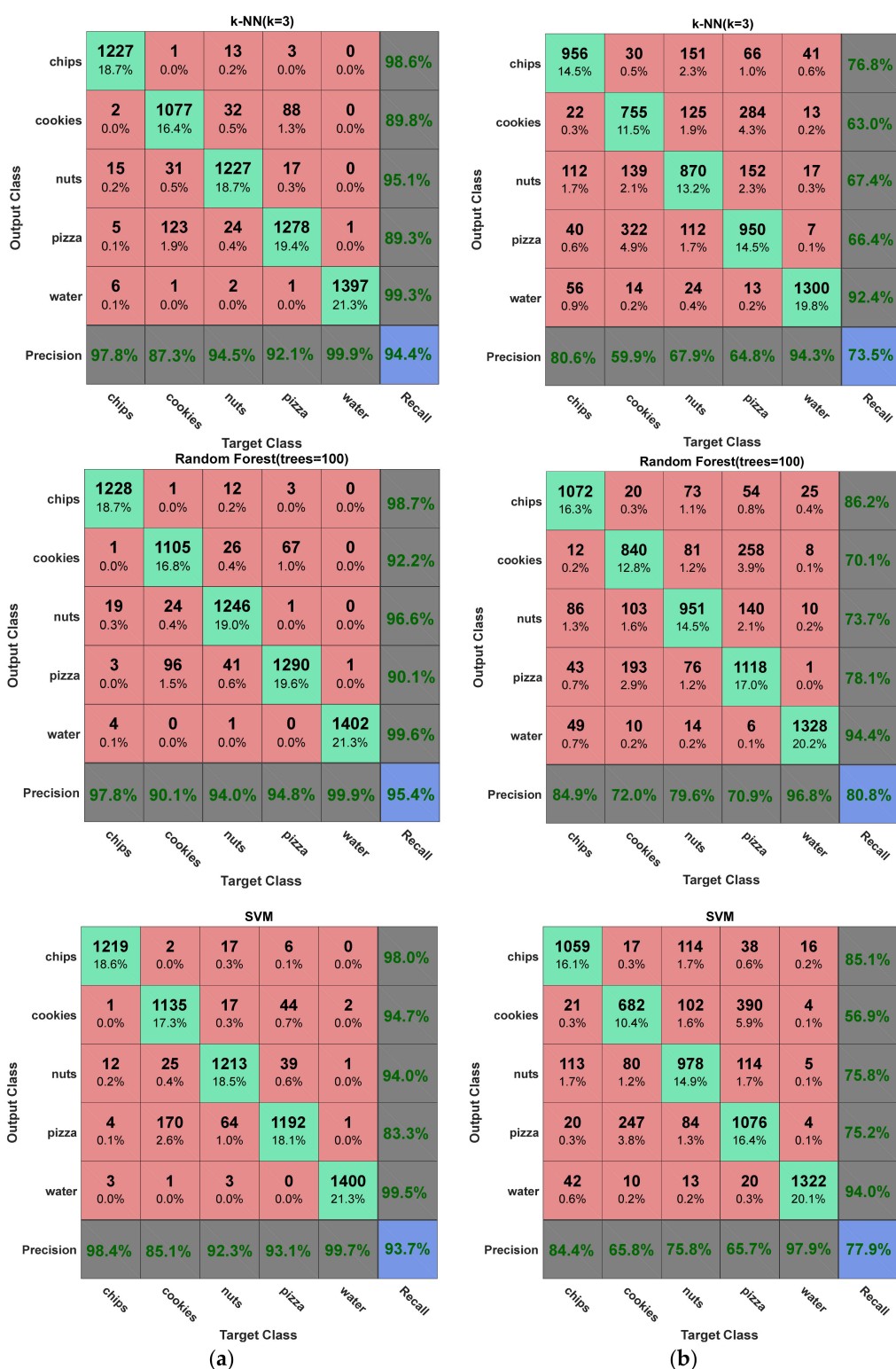

**Figure 8.** Food recognition performance: (**a**) Food recognition performance of the proposed method; (**b**) Food recognition performance of the previous method.

All three classifiers exhibited the highest classification for water over foods as water shares no common ingestion pattern with other food. Thus, the users' ingestion patterns for water were totally different, as can be seen in its generated waveform in Figure 4. The classifiers attained lower performance for cookies and pizza as instances belonging to these

foods were misclassified to each other. The misclassified or incorrectly classified foods may have common textural characteristics causing their incorrect classification.

### 6.2. Estimation of Food Weight and Caloric Intake

In our proposed method, the weight of food is measured from the number of swallows. The users were asked to swallow a bolus of food weighing about 1 g by taking a 1-g bite. Although lower than the natural food-bite size, this bite size helped the user to digest the food easily. Estimating the food weight based on the number of swallows is a novel idea and is simpler than previously employed food-weighing systems [26,27]. The estimation error was lower for foods with a discrete nature (chips, peanuts, and cookies) than for foods with a composite nature (pizza and water). Because the discrete foods were ingested easily by the users in appropriate bites, such foods could be estimated with high accuracy. In contrast, it remained very difficult for users to take 1-g sips of water or bites of pizza, so the errors for these items were large. Fortunately, water has zero calories, so it has no role in calorie estimation for the goal of weight control. The weight estimation using the proposed approach is shown in Figure 9. However, the calorie measurement estimation for pizza was degraded, as shown in Figure 10. The weight estimation for the other solid foods was highly accurate. High accuracy for weight estimates is important because it is the basis for caloric measurements.

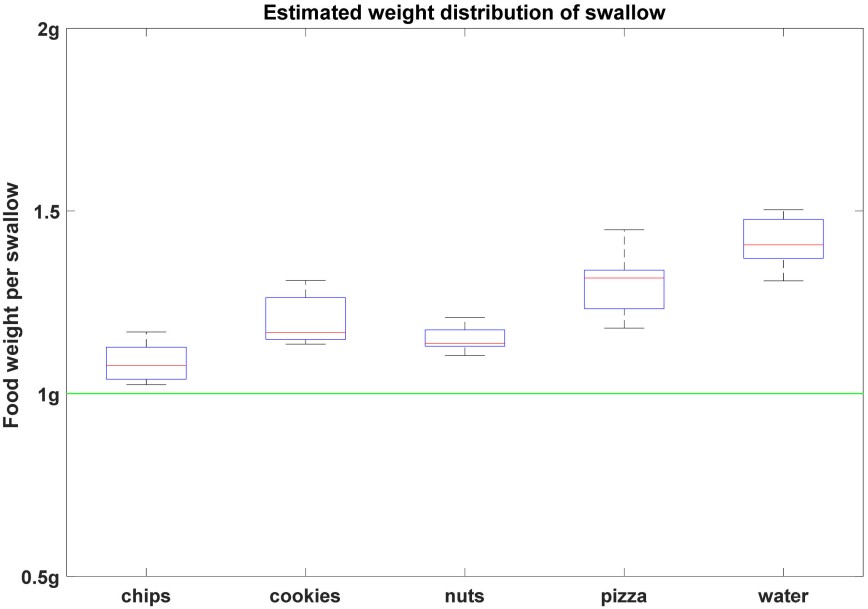

**Figure 9.** This figure describes the deviation of the estimated food weight from the actual weight of the consumed food.

Figure 9 displays the swallow-size comparison among different users for each food class. The horizontal line at 1 g represents the swallow-weight condition and acts as a reference for users over all foods; users were asked to take 1-g bites so that we could implement the stated strategy of weight estimation. Most of the foods shown in Figure 9 have short whiskers, indicating that all users swallowed similar amounts. However, pizza has large whiskers, indicating that different users swallowed different amounts. The estimated ingestion behavior of users exhibited error for swallow size, as graphically shown by boxes deviating from the reference line in Figure 9.

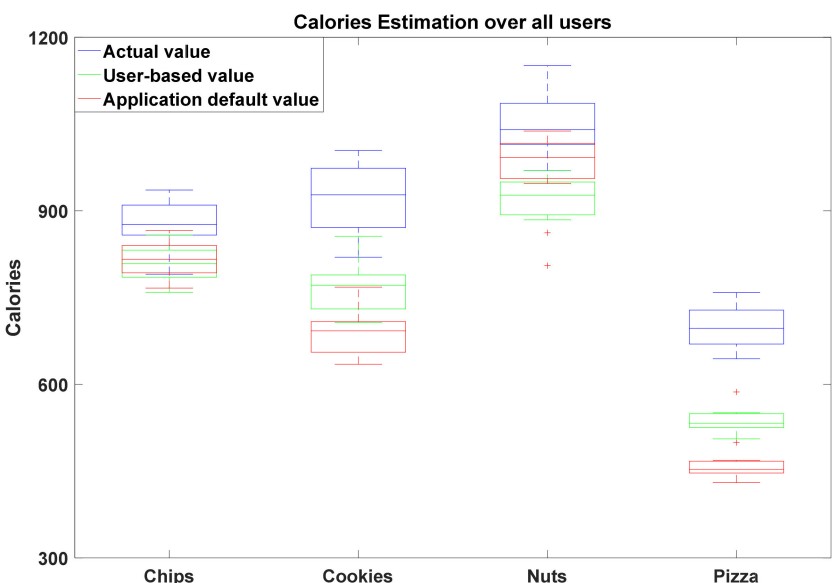

**Figure 10.** Calorie estimation methods and comparison with actual caloric intake.

Missed or undetected swallows and bites larger than 1 g were two sources of error in the estimation of food weight from swallow counts. Swallows greater than 1 g were the main contributor to the total error, while the remaining percentage of error was caused by missed swallows (21.3%, 13.4%, 12.5%, 16.1%, and 24.7% of the total error for chips, cookies, nuts, pizza, and water, respectively). For example, the overall error for water volume or weight was estimated as 29.26%, which can be further partitioned into 75.3% for large swallows and 24.7% for missed swallows. The percentage of error from missed swallows was smaller for the remaining foods, while the percentage of error from large swallow sizes was larger. All the participants in this experiment were young and were observed to take bites larger than the instructed 1 g for weight and calorie estimation. Thus, the weight and calorie estimation could be improved through careful selection of the amount of food for each bite.

The number of calories in each food is measured as the estimated weight of the food multiplied by its calorie weightage (the number of calories in 1 g of that food). As previously stated, the calorie weightage can be entered into the developed smartphone application in two ways: manually by the user, or automatically as the application default value, which is generalized over food varieties. The manual method requires a little effort from the user to enter the calorie value correctly and achieve the goal of calorie estimation. Alternatively, the user can avoid the extra burden of manually entering the calorie weightage by instead selecting the default value. For the automatic selection of calories, many varieties of the same food are considered, and the mean value is fixed as the default in the application (this was completed prior to the experiment). This method of calorie estimation can either minimize or maximize the error, depending on whether the default calorie weightage is higher or lower than the actual weightage. Figure 10 compares the relative performance of the two methods of estimating caloric intake. The whiskers of each box represent the calorie distribution and intra-person variability in calorie consumption for the related food. The differences in whisker height for boxes of different colors represent the errors that occurred in estimating the calories of the related food classes.

From this weight estimation, it is evident that calorie estimation error results from differences between the actual and estimated food weights. The estimated weight of food was observed to be less than the actual weight due to missed swallows and larger swallow sizes than the assumed 1 g. Clearly, error in the weight estimation is propagated in both methods of calorie estimation. Although the manual calorie estimation method requires the user to find out the calorie weightage of each food in advance, this method is only

subject to error if there are discrepancies in the weight estimation. In contrast, the calorie measurement based on the application default values is affected by both the propagated weight error and the generalized calorie weightage.

As shown in Figure 10, the manual method of calorie insertion was more accurate than the application default-based method for cookies and pizza. Because the application default-based method selected a smaller number of calories, the error difference increased as the gap between the actual and estimated calories increased. In contrast, the application-based method performed better in estimating the calories of chips and nuts because the default value was close to the actual value. The default calorie weightage clearly reduced the error gap caused by the weight estimation, so the calorie number was similar to the actual calorie number, as shown above in Figure 10. Thus, default values can regulate the error generated by weight differences, as they minimized the error for chips and nuts and maximized the error for cookies and pizza. The default calorie weightages of 5.25 for chips and 6.07 for nuts were higher than the actual values of 5.2 for chips and 5.67 for nuts. Similarly, smaller default values of 4.6 for cookies and 2.56 for pizza were chosen; the actual values are 5.125 for cookies and 3.01 for pizza. Thus, larger and smaller default values, reduced and increased the error, respectively, caused by weight discrepancy in the calorie estimation.

It has been graphically established that the application default weightage method has a higher error than the manual insertion method for some foods and a smaller error for other foods. Overall calories are calculated as the sum of the calories of all foods. Knowledge of the overall caloric intake can help the user to consume the required number of calories and to balance excessive calorie intake through physical exercise.

## 7. Conclusions

We have designed a wearable in the form of a necklace embedded with a piezoelectric sensor to monitor ingestion patterns for different foods. Our system extracts statistical features in the time domain and selects important features to enable the classifiers to recognize foods more accurately than the previous state-of-the-art piezoelectric sensor system that employs spectrogram features.

We have found that chewing patterns combined with a swallow event in the timedomain and a suitable number of samples in each frame are essential to accurately associate ingestion patterns to the relevant food category. Additionally, our system also exhibited the advantages of estimating the weights and calorie counts of solid foods. Our work has the limitation of including mainly on a young age group of subjects in the experiment. In the future, we aim to extend this system to monitor other activities across diverse age groups, to improve people's lifestyles and help them achieve the goal of better health. Moreover, determining an accurate caloric value may be possible with an advanced form of the food intake monitoring system in future studies.

**Author Contributions:** Conceptualization, G.H., B.A.S.A.-r., S.H., A.M.A., S.N.Q. and Z.A.; methodology, G.H., B.A.S.A.-r. and S.H.; software, G.H., B.A.S.A.-r. and S.H.; formal analysis and investigation, G.H., B.A.S.A.-r., S.H. and Z.A.; resources, Z.A.; writing—original draft preparation, G.H., B.A.S.A.-r., A.M.A., S.N.Q. and Z.A.; writing—review and editing, G.H., B.A.S.A.-r., A.M.A., S.N.Q. and Z.A.; project administration, A.M.A., S.N.Q. and Z.A. All authors have read and agreed to the published version of the manuscript.

**Funding:** The paper is funded by the Deanship of Scientific Research at Imam Mohammad Ibn Saud Islamic University Research Group no. RG-21-07-05.

**Informed Consent Statement:** Informed consent was obtained from all subjects involved in the study.

**Acknowledgments:** Authors extend their appreciation to the Deanship of Scientific Research at Imam Mohammad Ibn Saud Islamic University for funding this work through Research Group no. RG-21-07-05.

**Conflicts of Interest:** The authors declare no conflict of interest.

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
