# Peer review of "Smart Piezoelectric-Based Wearable System for Calorie Intake Estimation Using Machine Learning"

_applsci, doi:10.3390/app12126135_

Round 1
Reviewer 1 Report
The research presents a wearable 23 sensor in the form of a necklace embedded with a piezoelectric sensor, which detects skin movement 24 from the lower trachea while eating activity. After reading the manuscript, here are my comments:
- Motivation for the study must be given in the introduction section. What is the knowledge gap bridged by this study?
- The novelty of this study is not apparent. How the approach proposed by the authors differs from other similar approaches? Many papers published use machine learning for human activity recognition, including eating. What is your difference from the previous works and innovation? Clearly state the novelty of this study at the end of the introduction section.
- Research considers five food types typically using hand-based action. Why do you not choose spoon-based or others? Please, clarify.
- Swallow counts for each are not equal and varied. How does caloric intake acceptable?
- A performance comparison between the proposed study and previous related studies is required to show in detail (each activity).
There are some minor points needed to be corrected:
- Does OA denote Knee Osteoarthritis or Osteoarthritis?
- Font style in tables I and II.
- N. Alshurafa et al. or N. Alshurafa et al.?
- Fig. 8 or Figure 7.?
I consider that the authors need to address these issues before their paper can be published in the journal.
Author Response
Reviewer 1
The research presents a wearable 23 sensor in the form of a necklace embedded with a piezoelectric sensor, which detects skin movement 24 from the lower trachea while eating activity. After reading the manuscript, here are my comments:
- Motivation for the study must be given in the introduction section. What is the knowledge gap bridged by this study?
Response: Thank you for the comment. We have added the following paragraph in the introduction section as the motivation for designing a food intake monitoring system.
“Nowadays, the severity of chronic diseases has increased such that they become difficult to cure with traditional medicine. The medicine causes several severe side effects while curing a disease. Thus, several other diseases attacked the human body with traditional curing methods with medicine. Chronic diseases can be prevented if timely preventive steps are taken. Automated wearable monitoring systems are essential to assist users in following preventive measures. Henceforth, it motivated us to design a novel food intake monitoring system that can enable users to control their food intake calories and avoid deadly chronic diseases.”
2. The novelty of this study is not apparent. How the approach proposed by the authors differs from other similar approaches? Many papers published use machine learning for human activity recognition, including eating. What is your difference from the previous works and innovation? Clearly state the novelty of this study at the end of the introduction section.
Response: Thank you for pointing it out. It is already added on pages 2~3 of the revised manuscript.
“The contributions of our proposed food intake system are well described in three folds. First, we have achieved a higher classification accuracy than previous nutrition monitoring systems [2-7], [9-28]. We demonstrate that accurately processed raw temporal data along with an important set of optimal feature selection enables the proposed system to outperform the state-of-the-art necklace systems [10], [11] that use spectral features to recognize foods. We have observed that a temporal frame of 30 samples (1.5 seconds) containing a sequence of chewing events combined with a successive swallowing event forms a distinct pattern. The distinct temporal patterns in the frames generate essential effective features to associate the pattern with the relevant food category. Our proposed system has attained f-measures of 94.2%, 93.7%, and 95.1% using the k-nearest neighbor, Support Vector Machine, and Random Forest, respectively. A second important aspect of this algorithm is its high estimation rate for the weights of solid foods, the main contributor to caloric intake. To our knowledge, this is the first time that the weights of consumed foods have been estimated from the number of swallows. This method has achieved comparatively better recognition accuracy than the existing systems, with the additional advantage of avoiding restrictive or immovable models such as tables [26-27]. The estimated weight of food is converted into the caloric intake with sufficient accuracy to make users aware of their excessive caloric intake. Third, we have built a smartphone application that conveys the real-time notification about the number of swallows, an estimated food volume, and number of calories during each meal.”
3. Research considers five food types typically using hand-based action. Why do you not choose spoon-based or others? Please, clarify.
Response: Thank you for pointing out the eating supportive actions. In this study, our focus was to detect swallow counts and recognize food classes. However, we will add more sensors to the monitoring system for detecting the actions, which we think would further increase the accuracy of the designed system.
4. Swallow counts for each are not equal and varied. How does caloric intake acceptable?
Response: Swallow counts are different for each food class as they depend on amount of food ingested by the users. The food classes are not of same weight; therefore, the swallow counts are different. Caloric intake for each food is calculated using equation (1) given in the manuscript on page 7. Caloric value per swallow, denoted by c and different for each food, is multiplied with swallow counts of each food to compute food caloric value Ei. Then, caloric value for all foods is summed to find total caloric value E.
5. A performance comparison between the proposed study and previous related studies is required to show in detail (each activity).
Response: Thank you for valuable comment. The performance comparison of proposed study with the related studies is given in table 3 on page 14~15 of revised manuscript.
- There are some minor points needed to be corrected:
- Does OA denote Knee Osteoarthritis or Osteoarthritis?
- Font style in tables I and II.
- Alshurafa et al. or N. Alshurafa et al.?
- 8 or Figure 7.?
Response: Thank you for the comment.
- OA denotes Osteoarthritis
- Thank you for the correction.
- Thank you for the correction.
- It is Figure 8.
Reviewer 2 Report
This article proposes an innovative and improved detection system with wearable piezoelectric sensors, which can accurately classify and estimate food and weight by selecting the correct swallowing position to not only perform feature extraction on time-series chewing signals of appropriate duration, but also infer energy intake to reduce complexity and computational cost.
The whole paper is of good quality , however, there are still some issues in this paper:
- There are some pre-assumptions bringing the error between the calories calculated by the sensor and the actual calories of the food. The authors need further experiments and hypotheses closer to the real situation to demonstrate the practical value of this innovation by reducing the error of heat estimation through a more accurate algorithm.
- In addition, does the wearing of the necklace-like device affect the eating process? Authors, as researchers of auxiliary devices, need to be responsible for studying the relevant effects.
- Relying on chewing signals to classify food is not enough to deal with the multi-modality and variety of food in reality, and the authors need to clarify the use constraints of this innovation or replace a more realistic signal-to-calorie mapping relationship.
- Before the authors intercepts and analyzes the pictures of different signals caused by different foods (Figure 5), the authors need to describe how to distinguish chewing and swallowing in advance, rather than placing the description of chewing and swallowing signals after the conclusion.
- In Section 6.1 Food classification, the author uses SVM, Random forest and k-NN algorithms for classification. I am most concerned about which classification algorithm the author uses for his final innovation and how to use the features selected in Figure 7.
- The author needs to reduce the distance from evidence to conclusion. Existing images 4-8 are too far apart from the text of the quoted image. Author had better move figures to improve the correlation between the explanatory text and the picture where the evidence is located, and give all reader a good reading experience. Other pictures are as the same.
- Author’s work lacks of mathematical analysis. Could authors give some other mathematical evidence except for the experiments?
8.Some machine learning works are missed, [1] Deep-IRTarget: An Automatic Target Detector in Infrared Imagery using Dual-domain Feature Extraction and Allocation, [2] Graph-based few-shot learning with transformed feature propagation and optimal class allocation.
Author Response
Reviewer 2
This article proposes an innovative and improved detection system with wearable piezoelectric sensors, which can accurately classify and estimate food and weight by selecting the correct swallowing position to not only perform feature extraction on time-series chewing signals of appropriate duration, but also infer energy intake to reduce complexity and computational cost. The whole paper is of good quality, however, there are still some issues in this paper:
- There are some pre-assumptions bringing the error between the calories calculated by the sensor and the actual calories of the food. The authors need further experiments and hypotheses closer to the real situation to demonstrate the practical value of this innovation by reducing the error of heat estimation through a more accurate algorithm.
Response: Thank you for the comment. Missed or undetected swallows and swallows greater than 1-gram are two parameters contributing to the error. To the best of our knowledge, an error could be reduced with an advanced form of food intake monitoring system. It is one-of-the limitations of the proposed study, which is added in the conclusion section of a revised manuscript.
“Moreover, the accurate caloric value would be possible with an advanced form of food intake monitoring system in future studies.”
- In addition, does the wearing of the necklace-like device affect the eating process? Authors, as researchers of auxiliary devices, need to be responsible for studying the relevant effects.
Response: Thank you for the comment. There are two types of necklaces, reported in the research studies, for food intake monitoring. The necklace designs are pendants and sports bands. We chose the sports band design because it is suitable for experimentation as it does not affect the eating process.
The paragraph about necklace design is on page 5 of a revised manuscript.
“Piezoelectric sensors embedded necklace is designed in two different configurations: pendants and sports bands [13]. Unlike [4], [10-11], [19], we have selected the stretchable sports band design. The sports design can easily be stretched to the users’ neck, so people of different body structures can wear it without any extra effort, and hence it has better usability. As stated in [13], the stretchable necklace has the enormous advantages of comfortability and stability. The sports band is highly preferred in data collection, clinical environments, and algorithm evolution. The pendant [10-11], [19], the second necklace design, has the problem of leaving the neck skin position during activity motions such as vigorous head-turning, walking, or running.”
- Before the authors intercept and analyze the pictures of different signals caused by different foods (Figure 5), the authors need to describe how to distinguish chewing and swallowing in advance, rather than placing the description of chewing and swallowing signals after the conclusion.
Response: Thank you for the comment. A sequence of minor peaks denotes the chewing sequence and a large peak represents swallowing because a large force is exerted on the piezoelectric sensor embedded necklace while swallowing activity whereas chewing applies a smaller force. Henceforth, chewing activity generates smaller peaks, and swallowing causes large peaks.
The paragraph about necklace design is on page 8 of a revised manuscript.
“We combined the esophageal and pharyngeal phases to denote swallowing, and oral preparation phase to represent chewing. The main objective is to obtain the signals representing the users’ eating patterns for different foods. When food enters the mouth, different oral receptors are activated and carry sensory information about material properties to the brain. The most important stimuli are related to food texture (crispness, hardness, dryness, size, and shape) and flavor. The users continuously adjust these stimuli to efficiently break food into a bolus that can easily be swallowed. We have considered equal weightage for both actions: chewing and swallowing. As given in figure 4, the series of minor peaks represent chewing patterns and the major peaks with large amplitude represent swallowing patterns.
It is observed while ingesting different foods, users exert the different amounts of force during each attempt at food breakdown because foods vary in hardness, crispiness, and texture. Thus, the forces exerted to breakdown the foods with different levels of hardness generate distinct chewing patterns prior to each swallow. By exploiting the distinct ingestion patterns for foods with different textures, a series of chewing events are combined with swallowing events, which forms data of ingestion patterns. Unlike [10, 11], we have employed frames to borderline the desired ingestion pattern, ultimately enabling the efficient extraction of the efficient discriminating features. The frames enclosed in rectangular boxes are shown in Figure 4(b).”
- In Section 6.1 Food classification, the author uses SVM, Random Forest, and k-NN algorithms for classification. I am most concerned about which classification algorithm the author uses for his final innovation and how to use the features selected in Figure 7.
Response: Thank you for the comment. We have used three of baseline classifiers such as SVM, Random Forest, and k-NN for classification. Among the three of classifiers, random forest classifiers attained best performance as shown in figure 8 of revised manuscript. Relieff algorithm is applied on the manually extracted features to select most important discriminating features. Then, those discriminating features are fed into the classifiers to train for the classification task.
- The author needs to reduce the distance from evidence to conclusion. Existing images 4-8 are too far apart from the text of the quoted image. The author had better move figures to improve the correlation between the explanatory text and the picture where the evidence is located and give all reader a good reading experience. Other pictures are as the same.
Response: Thank you for the comment. It is fixed.
- Author’s work lacks of mathematical analysis. Could authors give some other mathematical evidence except for the experiments?
Response: Thank you for the comment. We have two equations in the manuscript. Equation (1) is used to compute total caloric value for all food classes. Caloric value for each food is calculated by multiplying ci (calorie value per gram) with the swallow counts of the food class i . Equation (2) is used for selecting the most essential discriminating features so that the classifiers can easily associate instances of food intake with particular food class.
- Some machine learning works are missed, [1] Deep-IRTarget: An Automatic Target Detector in Infrared Imagery using Dual-domain Feature Extraction and Allocation, [2] Graph-based few-shot learning with transformed feature propagation and optimal class allocation.
Response: Thank you for the comment. We added the suggested related studies in the revised manuscript on page 14.
“However, there are few studies based on advanced deep learning models that extract features and perform recognition together [35-36]. We have not used computationally complex deep learning models as we did not want to improve classification accuracy on the cost of high computation but preferred simple machine learning models. The exploited simple models attained better classification accuracy and can be integrated to wearable devices to perform operation in real-time.”
Round 2
Reviewer 1 Report
The authors have added some experimental results and modifications to respond positively to my questions in this version.
Reviewer 2 Report
Accept
This manuscript is a resubmission of an earlier submission. The following is a list of the peer review reports and author responses from that submission.